# A Korean Nationwide Cross-Sectional Study Investigating Risk Factors, Prevalence, and Characteristics of Sarcopenia in Men in Early Old Age

**DOI:** 10.3390/healthcare11212860

**Published:** 2023-10-30

**Authors:** Jongseok Hwang, Soonjee Park

**Affiliations:** 1Institute of Human Ecology, Yeungnam University, Gyeongsan-si 38541, Republic of Korea; sfcsfc44@naver.com; 2Department of Clothing and Fashion, Yeungnam University, Gyeongsan-si 38541, Republic of Korea

**Keywords:** sarcopenia, early old age, male, risk factors, prevalence

## Abstract

The present study investigated the risk factors, prevalence, and characteristics of sarcopenia among men aged 50–64 years. A total of 2868 participants were enrolled in this study. Of these, 328 individuals were classified into a sarcopenia group; the remaining 2540 were assigned to a control group. This study examined several variables, including skeletal muscle mass index, age, height, weight, body mass index, waist circumference, systolic and diastolic blood pressure, fasting glucose, triglyceride and total cholesterol levels, alcohol consumption, and tobacco use. It employed a stratified, clustered, and multistage probability sampling design. Complex sampling was used for the data analysis. The prevalence of sarcopenia was 10.25% (95% CI: 8.98–11.69). All anthropometric measures, including height, weight, BMI, and waist circumference, were significantly different between the two groups (*p* < 0.05). In terms of blood pressure, only systolic blood pressure (SBP) was significant (*p* < 0.05), and fasting glucose and triglyceride levels were risk factors for sarcopenia (*p* < 0.05). Tobacco use differed significantly between the two groups (*p* < 0.05). This study reported the specific prevalence of sarcopenia and identified its risk factors among men in early old age.

## 1. Introduction

Sarcopenia is characterized by the progressive loss of skeletal muscle mass, resulting in decreased muscle power, endurance, function, and overall quality of life [1]. In particular, skeletal muscle mass is one aspect of body composition, and is directly related with sarcopenia [2,3]. The exact mechanisms underlying sarcopenia are not yet fully understood; however, numerous studies have proposed several contributing factors, including hormonal changes, reduced physical activity, age-related alterations in muscle tissue, nutritional deficiencies, and neurodegenerative processes [4]. The decline in skeletal muscle mass typically begins around the age of 50, with an annual reduction of 1–2%, which further accelerates to 3% per year after the age of 65 [5].

The older adult population in Asia is rapidly growing, and Korea has the fastest aging rate globally. The Korean older adult population proportion was 16.5% in 2021 and is likely to increase to 39.8% by 2050 [6]. Consequently, compared to other countries, age-related conditions such as sarcopenia may have a negative impact on Korea and Asian countries.

Previous research has announced that the age-related decline in skeletal muscle mass is more common in males than in females [7,8,9,10]. Regardless of the considerable proportion of the elderly population at risk of developing sarcopenia and the noteworthy prevalence of sarcopenia in men, challenges remain in the identification of its risk factors and management within this population, particularly when compared to the extensive research available on sarcopenia in females [11,12,13,14]. Furthermore, Silva et al. [15] assessed 108 individuals with sarcopenia in 2021; however, they grouped both males and females together, which potentially limited their ability to identify sex-specific risk factors in males.

Some researchers conducted studies on elderly men with sarcopenia [16,17]. A study of Asian elderly patients with sarcopenia [17] investigated individuals aged 75 and above in 2022, and they identified several risk factors for sarcopenia, including height, weight, body mass index, waist circumference, skeletal muscle mass index, fasting glucose levels, and triglyceride levels. In 2022, Hwang and his colleague [16] focused on young old sarcopenic males aged 65 to 75, and found that age, body mass index, waist circumference, skeletal muscle index, fasting glucose levels, triglyceride levels, and systolic blood pressure were associated with the risk of sarcopenia. Hashmei [18] examined the factors contributing to sarcopenia among Iran’s elderly population in 2016. Their findings indicated that advancing age, smoking habits, and body mass index play roles as risk factors for the development of sarcopenia.

Likewise, most studies on sarcopenia have primarily concentrated on people aged 65 years and older [16,17,18,19,20,21]. Nevertheless, emerging research evidence advocates that age-related skeletal muscle loss begins at the age of 50 [22,23,24,25,26]. The occurrence of sarcopenia in early old age is attributed to a combination of complex factors. First, early-old-aged men experience hormonal alterations, such as reductions in growth hormone, insulin-like growth factor-1 levels, and testosterone. Growth hormone is produced by the pituitary gland and is essential for growth and tissue repair. It stimulates the growth and regeneration of cells, including muscle cells. IGF-1 is produced in the liver in response to growth hormone. It promotes the growth of various tissues. Higher levels of growth hormone are associated with increased muscle protein synthesis and muscle growth [27]. Testosterone, the primary male sex hormone, plays a critical role in enhancing muscle growth and strength. It boosts muscle protein synthesis, leading to larger and stronger muscles while also influencing the distribution of fat in the muscles [28]. These declining hormones level might be a contributing factor to the decline in muscle mass and strength observed during early old age [29,30]. Next, early-old-aged men frequently encounter difficulties in allocating time for consistent exercise routines [31]. This period typically coincides with the peak of their professional careers and their influential roles in society in their age group. They are often burdened with numerous responsibilities both at home and in their communities [32]. The pressing demands of their daily lives often lead them to deprioritize exercise routines, ultimately contributing to muscle loss as a consequence of insufficient physical activity. Moreover, they tend to opt for fast and convenient food options that are deficient in adequate nutrition [33]. This scarcity of essential nutrients, including protein, vitamins, and essential minerals [34], which are crucial for nurturing muscle health, also plays a substantial role in the premature onset of sarcopenia. Thus, sarcopenia in early old age arises from hormonal changes, reduced exercise opportunities, demanding life roles, and poor dietary choices, which all collectively influence sarcopenia onset.

Consequently, it is of utmost importance to identify the risk factors for muscle loss in men aged 50–64 years, as this knowledge can significantly contribute to the development of early prevention strategies for age-related muscle loss. Thus, the objective of this study was to examine specific clinical risk factors and their prevalence in men in early old age. This study formulated two hypotheses: (1) there is a specific prevalence of sarcopenia in the early-old-age group and (2) the exist distinct risk factors and characteristics associated with sarcopenia. 

## 2. Materials and Methods

### 2.1. Study Sampling

The Korea National Health and Nutrition Examination Survey (KNHANES) is an ongoing health surveillance program in South Korea designed to evaluate the well-being and dietary habits of the Korean population. It keeps track of health risk factors and the incidence of significant chronic diseases, and supplies vital data for shaping and assessing health policies and initiatives in the country. KNHANES is a comprehensive annual survey that covers the entire nation and is directed at a representative cross-section of the civilian population in South Korea. Each survey year features a fresh sample of approximately 10,000 individuals aged 1 year and older [35]. The Disease Control and Prevention Center’s ethics review board approved the experimental procedures of KNHANES, and all participants agreed to be involved in the study and signed a written informed consent form. The survey was conducted with 37,753 participants between 2008 and 2011. However, 34,528 individuals were excluded from our study as they were females, or males aged above 64 or below 50. This left a remaining sample of 3225 participants. An additional 357 participants were excluded because of an unavailability of health survey and DEXA measurement data. Consequently, the final analysis of data included 2868 participants who were assigned to two groups based on sarcopenia criteria. The sarcopenia group comprised 328 participants and the control group comprised 2540 individuals (Figure 1).

The study inclusion criteria included individuals who were male and aged from 50 to 64. Those excluded from the study were individuals lacking both DEXA measurements and health survey data, and anyone who had been hospitalized for any cause.

### 2.2. Sarcopenia Criteria

Sarcopenia is a medical condition classified under ICD-10-CM code M62.84, and its diagnosis involves the measurement of appendicular skeletal muscle mass. In this study, Dual X-ray absorptiometry (DEXA, QDR4500A, Hologic, Inc. Bedford, MA, USA) was employed to evaluate the appendicular skeletal muscle mass (ASM). The skeletal muscle mass index (SMI) was calculated by dividing the ASM (kg) by the body mass index (BMI) (kg/m^2^). Sarcopenia criteria are defined by the Foundation for the National Institutes of Health (NIH) Sarcopenia Project [36], where an SMI of less than 0.789 in men indicates the presence of sarcopenia. This diagnostic methodology was used to identify patients with sarcopenia.

### 2.3. Variables

#### 2.3.1. Anthropometric Variables

Anthropometric variables were gathered from all the participants. Before commencing the measurements, the participants were instructed to take off footwear, socks, and outerwear, such as coats, scarves, and hats, and wear lightweight clothing. Height and weight were evaluated using precise automated body measurement equipment, rounded to the nearest 0.1 cm or kg. BMI was calculated by dividing weight (kg) by the square of height (m^2^). Waist circumference (WC) was assessed during regular exhalation, with measurements taken to the nearest 0.1 cm on a horizontal plane at the midpoint between the lower rib and the upper edge of the hip bone (iliac crest). Standardized procedures were followed to ensure accurate and consistent anthropometric measurements in the study population [37].

#### 2.3.2. Blood Pressure

Systolic and diastolic blood pressure (SBP, DBP) were evaluated by a trained nurse using a mercury sphygmomanometer. During blood pressure measurement, the cuff was positioned at an equivalent level to the participant’s heart. Blood pressure was measured while the subject was sitting.

#### 2.3.3. Biochemical Variables

The biochemical variables measured in this study included the analysis and measurement of fasting glucose (FG), and triglyceride and total cholesterol (TC) levels. These tests were performed using a LABOSPECT 008AS platform manufactured by Hitachi High-Tech Co., Tokyo, Japan. Blood was collected from the nondominant arm after a fasting period of at least 8 h. Immediately after collection, the blood samples were mixed with a coagulation promoter and centrifuged. The above process was performed within 24 h of blood collection from the participants to ensure timely and accurate analysis of the blood samples.

#### 2.3.4. Tobacco Use and Alcohol Consumption

Information on tobacco use and alcohol consumption was obtained through surveys. The questionnaire was completed at a mobile examination center in a self-administered format by the subjects. Participants who reported smoking cigarettes and consuming alcohol were classified into three conditions: non-users, ex-users, or current users.

### 2.4. Data Analysis

Data are presented as mean ± standard deviation. The survey data were analyzed using a stratified, clustered, and multistage probability sampling design. Complex sampling analysis was conducted by adjusting the weights provided by National Health and Nutrition Examination Surveys in Korea, which enabled the sample to represent the entire nationwide Korean population. All statistical analyses were performed using SPSS (version 22.0; IBM Corp., Armonk, NY, USA). All data, including skeletal muscle index measured via DEXA, anthropometric measurements, blood pressure, biochemical variables, and tobacco and alcohol consumption, were sourced from the KNHANES dataset. Independent t-tests for parametric variables, including anthropometric measurements, blood pressure, and biochemical variables, and chi-square for non-parametric variables, such as tobacco and alcohol consumption, were used to compare variables between the two groups. Multiple logistic regression analysis was used to calculate the odds ratio for sarcopenia. A *p* value of <0.05 was considered statistically significant.

## 3. Results

### 3.1. Prevalence

The prevalence of sarcopenia is shown in Table 1 and Figure 2. The weighted prevalence was 10.25% (95% confidence interval (CI): 8.98–11.69). The un-weighted prevalence was 11.44%, and for the control group, was 88.56% (95% CI: 88.31–91.02).

### 3.2. Risk Factors

#### 3.2.1. Anthropometric Variables, SMI, and Age

Statistically significant differences were observed in height, weight, BMI, WC, SMI, and age (*p* < 0.05) (Table 2).

#### 3.2.2. Blood Pressure 

Significant differences were found in SBP (*p* < 0.05); DBP showed no significant differences (*p* > 0.05) (Table 3).

#### 3.2.3. Biochemical Variables

FG and triglycerides were significantly different between the two groups (*p* < 0.05), whereas TC did not differ significantly between the two groups (*p* > 0.05) (Table 3).

#### 3.2.4. Tobacco Use and Alcohol Consumption 

Tobacco use demonstrated a statistically significant difference (*p* < 0.05), whereas alcohol consumption was insignificant (*p* > 0.05). Table 4 shows the outcomes of tobacco and alcohol consumption.

### 3.3. Multiple Logistic Regression for Sarcopenia

Multiple logistic regression analysis was performed to evaluate the odds ratios for sarcopenia. Height, WC, SMI, SBP, FG, and triglycerides were statistically different from 1.0 (*p* < 0.05). The respective values were 47.000 (2.889–764.683), 0.010 (0.012–0.287), 1.476 (1.26–1.729), 38.059 (18.025–80.362), 1.082 (1.003–1.168), 1.511 (1.44–1.585), and 0.93 (0.911–0.948) (Table 5). The odds ratio values for variables such as height, WC, SMI, SBP, FG, and triglycerides were greater than 1.0, indicating a positive association. In contrast, the odds ratio for weight was less than 1.0, indicating a negative association.

## 4. Discussion

The present study investigated the incidence, clinical risk factors, and characteristics of sarcopenia in community-dwelling older adults. The aging population of Korea is rapidly increasing, resulting in a higher prevalence of sarcopenia, particularly in males. Healthcare professionals and clinicians face considerable challenges owing to limited knowledge and diagnostic tools, which hinder their ability to accurately identify and diagnose sarcopenia. Consequently, a significant number of cases of sarcopenia remain undetected, increasing the risk of potential complications. The study variables serve as cost-effective, convenient, and accessible indicators for detecting individuals at risk for sarcopenia. The variables examined included anthropometric measurements, blood pressure, blood laboratory test, alcohol consumption, and tobacco use. Understanding the risk factors associated with sarcopenia is crucial to enabling the early detection and implementation of preventive measures. The identified risk factors for sarcopenia were waist circumference, systolic blood pressure, fasting glucose, triglyceride levels, and tobacco use. In particular, the odds ratio associated with the outcomes indicates that taller height, decreased weight, enlarged waist circumference, enhanced skeletal muscle index, high systolic blood pressure, increased fasting glucose, and higher levels of triglycerides are more likely to occur in cases of sarcopenia.

WC has been reported as a risk factor. This finding is consistent with those of previous sarcopenic studies [7,38,39] on males. Brown et al. conducted a study on a U.S. cohort with an elderly population of 4425, where they found that WC was related to an increased odds ratio of 1.39 (95% CI: 1.05–1.84) for sarcopenia in males [7]. In addition, a sarcopenia study that assessed more than 600 Brazilian people living in urban area revealed that the sarcopenic group had increased WC compared to the healthy group. [38]. Another Japanese cross-sectional study conducted by Sanada on 1488 non-institutional adults aged 18–85 reported that people with sarcopenia had high WC compared to the healthy population [39]. A plausible underlying reason for higher WC as a risk factor is the reciprocal relationship between increased fat mass and decreased muscle mass [40]. Individuals with sarcopenia often experience decreased muscle power and function due to muscle loss, leading to reduced levels of physical activity [41]. A decline in the level of physical activity is correlated with diminished total daily energy expenditure and bigger fat deposition, especially in the visceral and abdominal areas, thereby contributing to increased waist volume [41]. Furthermore, a higher fat volume, especially visceral fat, promotes the release of pro-inflammatory cytokines, including interleukin 6 and C-reactive protein, which might impede the anabolic response of muscle tissue [42]. Consequently, the relationship between reduced muscle mass and increased fat mass in patients with sarcopenia is interdependent [43].

Fasting glucose was a risk factor for sarcopenia. This study is parallel with previous research [44,45]. In 2018, Du et al. conducted a meta-analysis to assess the differences in metabolic risk factors between patients with sarcopenia and the healthy population, and disclosed that glucose was risk factor for sarcopenia [44]. Ozturk carried out a study on sarcopenia in older men in 2018, specifically in the Central Asia region. The study involved 63 participants with an average age of 70 years and found that sarcopenic patients had high blood glucose [45]. A possible underlying mechanism for such high blood glucose in sarcopenic participants is the role of skeletal muscle mass in blood glucose regulation [46] Glucose in lean muscle mass blood is removed via vasodilation and perfusion; then, lean mass blood flow and perfusion cause vasodilation and the process triggers glucose clearance [47]. Up to eight percent of glucose is stored in skeletal muscle mass after a meal. [48]. A lack of muscle mass, such as in sarcopenia, leads to less storage of glucose and less glucose remaining in the blood [45,49]. Furthermore, insulin resistance, frequently accompanied by hyperglycemia, may lead to muscle loss. Insulin resistance reduces protein synthesis and increases protein degradation by deactivating the mammalian target of rapamycin [50]. The relationship between skeletal muscle loss and increased levels of fasting glucose is bidirectional and mutually reinforcing.

Triglyceride levels were also identified as a risk factor for sarcopenia, with 201.201 mg/dL in the sarcopenia group compared to 172.042 mg/dL in the non-sarcopenia group. This outcome is in line with that of previous sarcopenia research [51,52,53]. Lu et al. [53] conducted a study of 600 older adults residing in northern Taiwan, and reported that the sarcopenia group had significantly higher triglyceride levels (1.9 mmol/L) than the control group (1.3 mmol/L). Similarly, Buchmann et al. [51] investigated 1420 older adults living in Berlin and found that triglyceride levels were higher in the sarcopenia group compared to the healthy group; its respective values were 108.7 mg/dL and 92.1 mg/dL. An Eastern China study [52] observed increased serum triglyceride levels in males with sarcopenia. One possible explanation for elevated triglyceride levels is the presence of insulin resistance [54] and the release of inflammatory cytokines [55].

Tobacco use was also identified as an additional risk factor. This finding is consistent with those of previous studies investigating the risk factors for sarcopenia [56,57]. A U.S. study on sarcopenia including 11,616 adult participants from the National Health and Nutrition Examination Survey reported that a higher percentage of individuals in the sarcopenia group had a history of tobacco use than the normal population [56]. A British cohort study of older adults with sarcopenia demonstrated that smoking was related to the risk of sarcopenia compared to the healthy older adult population [57]. A plausible theoretical rationale for smoking status as a risk factor for sarcopenia is as follows. Cigarette smoking exacerbates the decline in skeletal muscle mass by inhibiting muscle protein synthesis, accellerating muscle breakdown. Smoking is associated with a decrease in the fractional synthesis rate of muscles and alterations in genes related to muscle atrophy and muscle growth inhibition, including E3 ubiquitin ligase muscle atrophy [58]. Smokers have been found to exhibit a loss of type I muscle fibers, enhanced glycolytic capacity, and diminished expression of constitutive nitric oxide synthases, which contribute to a decrease in skeletal muscle volume [59]. In particular, male smokers have lower testosterone levels than non-smokers [60,61]. This is due to dysfunction in Leydig cells in male smokers. Leydig cells are the primary source of testosterone and are damaged by mechanisms such as chronic hypoxia, smoke-induced oxidative stress, and the neuroendocrine effects of nicotine on the hypothalamic–pituitary–gonadal axis [62]. These mechanisms result in impaired Leydig cell function, the disruption of testosterone production, and negative effects on muscle health. These factors cause Leydig cell dysfunction and disrupt testosterone production. Reduced blood testosterone levels have a negative effect on muscle synthesis in males [63].

SBP was a notable risk factor for sarcopenia in men. These outcomes are consistent with previous studies [53,56,57,64]. In a cohort study by Atkins et al. [57], which involved 4252 participants from Britain, those with sarcopenia exhibited greater SBP than those without. Similarly, Androga et al.’s [56] cohort study in the U.S. reported a higher prevalence of hypertension among individuals in the sarcopenia group than among the control group. Additionally, Yin et al. [64] assessed approximately 15,000 Chinese individuals and found that men with sarcopenia displayed elevated SBP compared with their healthy counterparts. Several potential mechanisms may account for the fact that a higher SBP is a clinical risk factor for sarcopenia in men. Metabolic alterations and muscle mass loss occur owing to reduced energy expenditure and physical activity. These changes may contribute to insulin resistance and arterial stiffness [65,66,67]. Specifically, excess calories owing to reduced energy expenditure may accelerate fat accumulation in the visceral area. The accumulation of visceral fat mass is able to induce an inflammatory response, which may lead to the thickening of blood vessel walls, the constriction of vascular passages, and the subsequent impairment of blood flow [68].

The strength of the present study lies in its examination of a representative Korean population of early-old-aged adults in whom skeletal muscle loss had begun. This approach is distinct from many studies that have combined both sexes into a single group [4,14,53]. However, this study has a couple of limitations that should be considered in future research. First, cross-sectional studies have the capacity to discern associations between risk factors and outcomes, but they may lack comprehensive control over confounding variables, which encompass factors capable of exerting influence on both the risk factor and the outcome in question. To enhance the robustness of the findings, future studies should incorporate longitudinal or randomized case–control designs. Second, this study did not include sarcopenic obesity. Considering sarcopenic obesity conditions would provide a better understanding of fasting glucose levels. In addition, this study did not address risk factors such as nutrition and physical activity, even though these factors are well documented. Incorporating an analysis of these factors would have enhanced the quality of this paper. Finally, our research did not provide more comprehensive comparisons between large cities and small towns, and urban and rural areas, while also considering diverse age groups. To bolster the quality of future studies, they should carefully consider and incorporate these above limitations.

## 5. Conclusions

This study provides the first clinical evidence of the risk factors, prevalence, and characteristics of sarcopenia among men in early old age in Korea. The sarcopenia prevalence in weighted value was 10.25% (95% CI: 8.98–11.69). The risk factors included height, weight, BMI, WC, SMI, FC, triglycerides, SBP, and tobacco use. Healthcare providers and clinicians may enhance their ability to identify and detect male patients with sarcopenia by understanding its prevalence and associated risk factors. To further strengthen the reliability of our findings, future research should consider employing longitudinal or randomized case–control study designs.

## Figures and Tables

**Figure 1 healthcare-11-02860-f001:**
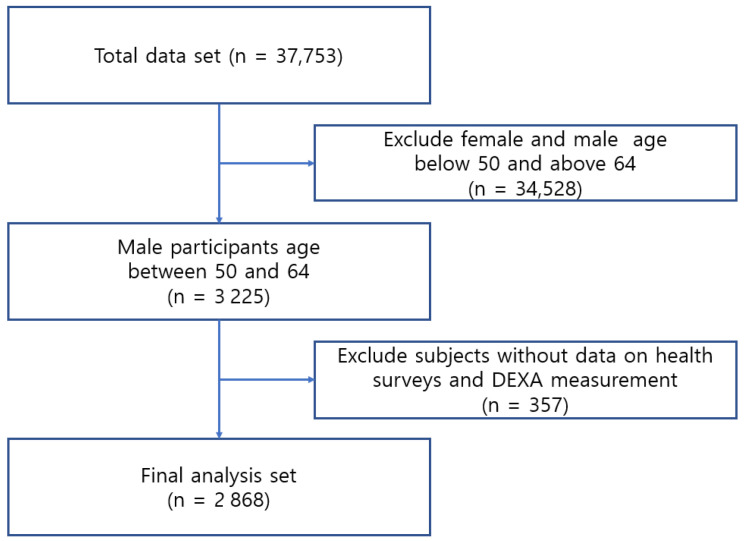
Flowchart for subject selection.

**Figure 2 healthcare-11-02860-f002:**
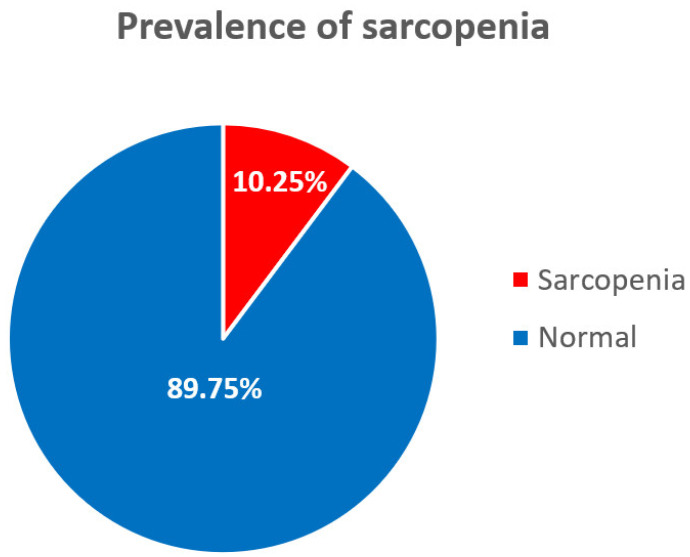
Diagram of prevalence of sarcopenia.

**Table 1 healthcare-11-02860-t001:** Prevalence of sarcopenia.

	Sarcopenia	Normal	Total
	(n = 328)	(n = 2540)	(n = 2868)
Un-weighted (%)	11.44	88.56	100
Weighted (%)	10.25 (8.98–11.69)	89.75 (88.31–91.02)	100

Weighted values expressed with 95% confidence interval.

**Table 2 healthcare-11-02860-t002:** Age, anthropometric variables, and SMI.

	Sarcopenia	Normal	*p*
	(n = 328)	(n = 2540)	
Age (years)	56.6 ± 5.4	53.9 ± 5.7	<0.001
Height (cm)	161.4 ± 5.1	169.2 ± 5.2	<0.001
Weight (kg)	67.4 ± 9.6	68.8 ± 9.5	<0.001
BMI (kg/m^2^)	25.7 ± 3.0	23.9 ± 2.8	<0.001
WC (cm)	89.0 ± 8.3	85.1 ± 8.2	<0.001
SMI (kg/m^2^)	742.4 ± 41.1	926.3 ± 84.8	<0.001

Values are presented as mean ± standard deviation. The independent t-test was used. BMI, body mass index; WC, waist circumference; SMI, skeletal muscle index.

**Table 3 healthcare-11-02860-t003:** Blood pressure and biochemical variables.

	Sarcopenia	Normal	*p*
	(n = 328)	(n = 2540)	
SBP (mmHg)	126.9 ± 15.7	123.9 ± 16.4	0.002
DBP (mmHg)	81.6 ± 10.5	81.8 ± 10.6	0.734
FG (mg/dL)	111.3 ± 37.0	103.9 ± 26.7	<0.001
Triglycerides (mg/dL)	201.2 ± 151.1	172.0 ± 156.9	0.002
TC (mg/dL)	191.5 ± 40.8	191.5 ± 36.4	0.974

Values are presented as mean ± standard deviation. The independent t-test was used. SBP, systolic blood pressure; DBP, diastolic blood pressure.; FG, fasting glucose; TC, total cholesterol.

**Table 4 healthcare-11-02860-t004:** Tobacco use and alcohol consumption.

	Sarcopenia	Normal	*p*
	(n = 328)	(n = 2540)	
Tobacco use (%) (current/ex-/non-user)	64.4/19.3/16.2	56.9/30.2/12.8	0.001
Alcohol consumption (%) (current/ex-/non-user)	80.5/12.3/7.0	84.7/10.5/4.7	0.175

The chi-square test was used.

**Table 5 healthcare-11-02860-t005:** Odds ratios of sarcopenia.

Variables	Odds Ratio (95% of CI)	*p*
Height	47.0 (2.8–764.6)	0.007
Weight	0.0 (0.0–0.2)	0.007
WC	1.4 (1.2–1.7)	<0.001
SMI	38.0 (18.0–80.3)	<0.001
SBP	1.0 (1.0–1.1)	0.043
FG	1.5 (1.4–1.5)	<0.001
Triglycerides	1.0 (1.0–1.1)	<0.001

All values represent multiple logistic regressions with 95% confidence intervals (CI). BMI, body mass index; WC, waist circumference; SMI, skeletal muscle index; SBP, systolic blood pressure; FG, fasting glucose.

## Data Availability

All data were anonymized and can be downloaded from the website at https://knhanes.kdca.go.kr/knhanes, accessed on 1 October 2023.

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
