# Peer review of "A Korean Nationwide Cross-Sectional Study Investigating Risk Factors, Prevalence, and Characteristics of Sarcopenia in Men in Early Old Age"

_healthcare, 2023, doi:10.3390/healthcare11212860_

Round 1
Reviewer 1 Report
Comments and Suggestions for Authors
Thank you for submitting your work, it was of interest. I have a few comments that should be addressed:
- Methods - I am slightly unclear about where certain pieces of data came from in the study. You mention that data from the KNHANES survey was used, however, it was not clear where the DEXA measurement data came from or any of the body measures. Further detail on how data analysed in this study was collected.
- Results/Discussion - I was slightly confused by your interpretation of the odds ratio analysis presented in the results section. You state that various measures showed statistical differences in the multiple logistic regression due to the p value being less than 0.05. However, the p value in this analysis allows you to determine whether the odds ratio value is statistically different from 1 and therefore whether there is a relationship between the variable in question and the events probability. I think you need to be clearer when presenting the results in this section and then how you interpret the results in your discussion.
Comments on the Quality of English LanguageThere are quite a few instances of poor written English in this manuscript which would need to be reviewed.
Author Response
At first, authors express their deep gratitude to the reviewer’s valuable comments. know it is arduous job for reviewing papers. It consumes lots of time and effort. We were able to learn the way to write paper correctly because of your delicate comments. And we came to realize citing new references is very important, again. We really appreciate it with my whole heart.
*Please find attached our revised manuscript with changed from the original version highlighted with turquoise color (Please click an author's note file).

Reviewer 2 Report
Comments and Suggestions for Authors
Thank you for submitting the healthcare application. And it is an honor to review it.
It includes Nationwide in the title: I think it should include a broad and diverse analysis, such as differences between big cities and small towns, or cities and rural areas, or differences by age group, or between men and women.
It includes Risk Factors in the title. The risk factors for Sarcopenia are very diverse. and nutrition and physical activity. Unfortunately, this information is not available.
Line 170: Title needs to be corrected.
Line 165: interval
0.000 to <0.001
Combine Tables 3 and 4 into one.
Combine Tables 2 and 5.
The decimal points in Tables 2, 3, 4, and 5 are unified to 1 digit. (P value maintains 3 digits)
Comments on the Quality of English LanguageThere is a problem with writing academic documents rather than a grammatical problem.
Author Response
At first, authors express their deep gratitude to the reviewer’s valuable comments. know it is arduous job for reviewing papers. It consumes lots of time and effort. We were able to learn the way to write paper correctly because of your delicate comments. And we came to realize citing new references is very important, again. We really appreciate it with my whole heart.
*Please find attached our revised manuscript with changed from the original version highlighted with green (Please click an author's note file).

Reviewer 3 Report
Comments and Suggestions for Authors
Please consider the following points:
-Introduction:
The introduction provides a comprehensive overview of sarcopenia and its contributing factors, as well as the increasing prevalence of sarcopenia in the older adult population in Korea and Asia. However, there are a few areas that could be improved:
1. Lack of citation: The introduction mentions several studies and statistics without providing proper citations. It is important to include references for the information presented to support the claims made. especially you should provide some information about body composition and then sarcopenia. you can use the following references:
*Nabilpour, Maghsoud, and Jerry Mayhew. "Effect of peripheral heart action on body composition and blood pressure in women with high blood pressure." International Journal of Sport Studies for Health 1.2 (2018).
Irandoust, Khadijeh, and Morteza Taheri. "The effects of aquatic exercise on body composition and nonspecific low back pain in elderly males." Journal of physical therapy science 27.2 (2015): 433-435.
2. Organization: The introduction jumps between different topics and studies without clear transitions. It would be beneficial to reorganize the information to provide a more logical flow of ideas.
3. Repetition: Some information is repeated unnecessarily, such as the mention of the prevalence of sarcopenia in men compared to women in multiple studies. This repetition could be avoided to maintain a more concise and focused introduction.
4. Lack of clarity: The introduction briefly mentions that middle-aged men experience hormonal alterations, but it does not clearly explain how these alterations contribute to sarcopenia. Providing more specific details and explanations would enhance the clarity of the introduction.
5. Inconsistency in referencing previous studies: The introduction mentions previous studies on sarcopenia but does not consistently provide the author names or publication dates. Including this information would make it easier for readers to locate and reference the cited studies.
Overall, with some revisions and improvements in organization, citation, and clarity, the introduction could effectively introduce the topic of sarcopenia and its relevance to the Korean and Asian populations.
-The Materials and Methods section provides a detailed description of the study sampling and variables measured. However, there are a few areas that could be improved:
1. Lack of explanation: The section jumps straight into describing the Korea National Health and Nutrition Examination Survey (KNHANES) without providing any context or rationale for why this survey was chosen for the study. I
2. Lack of clarity: The section mentions that 35,737 participants were included in the survey between 2008 and 2011, but it is not clear how these participants were selected or what criteria were used for inclusion.
3. Lack of information: The section mentions that 3,225 participants were included in the final analysis, but it does not explain how these participants were selected from the initial sample.
4. Inconsistency in referencing: The section mentions the Foundation for the National Institutes of Health (NIH) Sarcopenia Project as the source for defining sarcopenia criteria, but it does not provide any citation or reference for this project.
5. Lack of detail: The section briefly mentions that anthropometric variables, blood pressure, biochemical variables, tobacco use, and alcohol consumption were measured, but it does not provide any specific details about how these measurements were conducted or analyzed.
Overall, with some revisions and improvements in clarity, detail, and referencing, the Materials and Methods section could effectively describe the study sampling and variables measured.
-Results
*its better to use figure(s) to show the results
-Discussion
Overall, the discussion provides a clear and concise summary of the study findings and their implications. The authors effectively relate their results to previous research studies, highlighting the consistency of their findings with existing literature. They also provide plausible explanations for the observed associations between the risk factors and sarcopenia. However, there are a few areas that could be improved:
1. Lack of detail: While the discussion briefly mentions previous studies that have reported waist circumference and fasting glucose as risk factors for sarcopenia, it does not provide specific details about these studies (e.g., sample size, methodology, findings). Including more specific information about these studies would strengthen the discussion and provide a more comprehensive understanding of the existing literature.
2. Lack of discussion on other risk factors: The discussion focuses solely on waist circumference and fasting glucose as risk factors for sarcopenia, but it does not mention or discuss other variables measured in the study (e.g., systolic blood pressure, triglyceride levels, tobacco use). Providing a brief discussion of these variables and their potential implications for sarcopenia would enhance the completeness of the discussion.
3. Limited discussion of limitations: The discussion does not address any limitations of the study or potential sources of bias that may have influenced the results. Including a brief discussion of limitations would acknowledge any potential weaknesses in the study design or methodology and provide a more balanced interpretation of the findings.
Author Response
At first, authors express their deep gratitude to the reviewer’s valuable comments. We know it is arduous job for review papers. It consumes lots of time and effort. We were able to learn the way to write paper correctly because of your delicate comments. We really appreciate it with my whole heart.
*Please find attached our revised manuscript with changed from the original version highlighted with Yellow (Please click an author's note file).

Round 2
Reviewer 2 Report
Comments and Suggestions for Authors
Many parts have been revised. I hope you have good results.
Comments on the Quality of English LanguageNo special comments
Reviewer 3 Report
Comments and Suggestions for Authors
the revisions are accepted.